# On the Achievable Capacity of MIMO-OFDM Systems in the CathLab Environment

**DOI:** 10.3390/s20030938

**Published:** 2020-02-10

**Authors:** João Guerreiro, Rui Dinis, Luís Campos

**Affiliations:** 1IT, Instituto de Telecomunicações, 1049-001 Lisboa, Portugal; rdinis@fct.unl.pt; 2PDMFC, Projecto Desenvolvimento Manutenção Formação e Consultadoria LDA, 1300-609 Lisboa, Portugal; luis.campos@pdmfc.com; 3FCT-UNL, Universidade Nova de Lisboa, Universidade Nova de Lisboa, Monte de Caparica, 2829-516 Caparica, Portugal

**Keywords:** medical imaging, propagation, MIMO, capacity, OFDM

## Abstract

In the last years, the evolution of digital communications has been harnessed by medical applications. In that context, wireless communications are preferable over wired communications, as they facilitate the work of health technicians by reducing cabling on the stretchers. However, the use of wireless communications is challenging, especially when high data rates and low latencies are required. In those scenarios, multiple-input multiple-output (MIMO) techniques might have an important role, thanks to the high capacity gains that they can exhibit, which ideally increase with the MIMO size. In this work, we study the propagation scenario of a typical medical laboratory through ray-tracing techniques. By taking into account the derived channel model, we study the potential of MIMO techniques in an IEEE 802.11ax environment. Through a set of performance results regarding the system capacity, we show that the MIMO gains might not be as high as supposed in the medical laboratory, being far from the ideal scenario. Therefore, the large data rates required by the modern medical imaging applications might only be achieved with a combination of MIMO systems and large bandwidths.

## 1. Introduction

Medical imaging procedures have been evolving substantially in the last years. An example of this evolution observed nowadays is the catheter imaging procedures, which allow the catheter to transfer a high-resolution image with a low latency to an external receiver, using intravascular ultrasound (IVUS) techniques [1]. This advanced procedure was enabled by using evolved materials, miniaturized electronics, and digital communications [2]. With regard to digital communications, current catheter imaging is based on wired communications [3,4]. However, the use of wired communications can complicate the physician’s task due to the high number of cables in his work area, which can even reduce the quality and the results of the procedure, not to mention possible sterilization problems. In that context, the use of wireless communications would be greatly preferable [5].

The design of wireless communications is very challenging, especially in applications such as catheter imaging, where it is important to combine very high data rates with low latencies. When it comes to the required data rates, the typical values range from approximately 300 Mbps to over 1 Gbps. Moreover, a eye–hand coordination is desired, which means that the latency should not be larger than 100 ms. Theoretically, these strict radio requirements can be obtained with the recent wireless local area network (WLAN) communication standard IEEE 802.11ax (also known as WiFi 6) [6], which announces several improvements in terms of data rates relatively to its antecessor, the so-called IEEE 802.11ac [7]. However, even in the IEEE 802.11ax, these data rates are only achievable with large bandwidths and/or with multiple-input, multiple-output (MIMO) techniques [8]. In fact, it is widely known that the use of MIMO schemes can lead to substantial gains in terms of system capacity and/or reliability. However, these gains only take place if the propagation channel is rich in terms of separable multipath components [9]. Indeed, to take advantage of the full MIMO potentialities, the MIMO channels should have a large number of relevant singular values [10]. If this is not the case, the achievable MIMO gains can be well below the theoretical values [11]. In those conditions, the application of MIMO techniques should be carefully considered, as the additional gains might not compensate for the increased hardware complexity and for the substantial increase in the signal processing of the transceivers.

To evaluate the viability of MIMO techniques in wireless medical applications, we present a channel model of a conventional catheter laboratory (CathLab), which is obtained by considering the propagation characteristics of the medium and through the use of ray-tracing techniques. By combining this propagation model and an orthogonal frequency division multiplexing (OFDM) [12] of an IEEE 802.11ax system, we obtained several results regarding the system capacity. We show that the true MIMO gains in the CathLab are far away from the theoretical ones. This can be explained by the low multipath effects (i.e., the poor scattering environment), which might lead to correlated channels and ill-conditioned channel matrices. Therefore, the use of MIMO techniques with a large number of antennas based on the WLAN IEEE 802.11ax standard might not be a good solution for achieving very high data rates in wireless medical imaging applications.

This paper is organized as follows. In Section 2, the CathLab and the 802.11ax system are characterized. Section 3 is dedicated to modeling of the CathLab channel. Section 4 concerns with the channel capacity of the CathLab. Section 5 concludes this work.

## 2. System Characterization

Consider Figure 1, which represents an example of a medical imaging laboratory (in this case, the Philips CathLab^®^).

In the CathLab, we have a patient on a stretcher and several medical staff (cardiologist, nurses, anesthetist, etc.). The doctor inserts an IVUS catheter in the patient, and this IVUS catheter is connected to a patient interface module (PIM), which transmits a sequence of images to a receiver through a wireless communication. The wireless communication between the PIM, which we can assume to be in doctor’s hand, and the receiver, which is installed somewhere in the stretcher, or close to it, is made through a WLAN IEEE 802.11ax system operating with a carrier frequency $f_{c}=5.1$ GHz. For this scenario, we considered a MIMO-OFDM system where the PIM (i.e., the so-called station (STA) or transmitter) is equipped with *M* transmit antennas and receiver (i.e., the so-called access-point (AP)) is equipped with *N* receive antennas. The theoretical achievable data rate *R* of the IEEE 802.11ax connection is dependent on (i) the channel bandwidth *B*, which can be obtained from a single channel or by aggregation of two channels; (ii) the MIMO size (i.e., the values of *M* and *N*); and (iii) the adopted modulation and coding scheme (MCS), which defines both the modulation and the code rate (CR). Table 1 shows the theoretical achievable data rates per spatial stream considering different MCS and channel bandwidths [13].

Note that there are different combinations of MCSs and bandwidths that can yield data rates from approximately 600 Mbps to over 1Gbps, but all require the use of large bandwidths (B>40 MHz). However, it can be difficult to obtain such bandwidths, as the maximum “guaranteed” bandwidth is 40 MHz and bandwidths larger than this are only obtained if they are idle (i.e., not used by other STAs). This can lead to significant problems in terms of quality of service (QoS) [14], as the available bandwidth for TX transmission would be variable and dependent on the success of the medium access. This is not tolerable in the CathLab, where the QoS requirements are very strict, and where one cannot accommodate variations on the quality of the image transferred by the IVUS. Under these conditions, the adoption of more than one spatial stream (i.e., the adoption of a MIMO scheme) can be a solution to increase the achievable data rates without the need for a large bandwidth. At a first glance, it would be expected that the achievable rate with *S* spatial streams is *S* times the data rate associated to one spatial stream. However, this only holds for Rayleigh fading channels, where it is assumed that channels between different antennas are uncorrelated. In practice, to access the system capacity, the CathLab channel should be modeled, so that the potential MIMO gains can be evaluated.

## 3. CathLab Channel

### 3.1. Ray-Tracing

In this subsection we analyze the propagation channel of the CathLab through a 2-D simulation based on ray-tracing techniques [15,16]. The main goal is to derive channel impulsive response (CIR) and the channel frequency response of the CathLab.

As can be seen in Figure 2, we consider an x–y coordinate system where a given point $p_{h}$ is defined as $p_{h}=\left(x_{h},y_{h}\right)$. We consider that the CathLab room is positioned at the origin of this system, i.e., $p_{c}=\left(x_{c},y_{c}\right)=\left(0,0\right)$ m. The CathLab dimensions in terms of length and width are $c_{length}$ and $c_{width}$, respectively. The position of the stretcher is defined as $p_{s}=\left(x_{s},y_{s}\right)$, and the dimensions of the stretcher are $s_{length}$ and $s_{width}$. Figure 4 shows the top view of the CathLab, together with the definition and positioning of the room and the stretcher.

Unless otherwise stated, we consider that the dimensions of the CathLab are $c_{length}=6$ m and $c_{width}=4.5$ m. In fact, although the ceiling and floor can add some additional multipath components, for the sake of simplicity, we are not considering the height of the room. The validity of this simplification can be explained by the fact that we almost never have a ground reflected component since those components are usually blocked by the stretcher (i.e., the corresponding reflecting point would be at the stretcher, not at the floor). With respect to the ceiling, the radiation pattern of the receive antenna (that is usually in a fixed position) can reduce substantially the power of multipath components coming from above (i.e., it is designed to be almost omnidirectional in the horizontal plane, but it has some directivity in the azimuth). Moreover, in many cases, the reflecting properties of the ceiling material are not as good as the ones of the walls.

We consider that the stretcher position is $p_{s}=\left(2,1\right)$ m and that the stretcher dimensions are $s_{length}=2$ m and $s_{width}=0.5$ m. We also assume that the position and orientation of the array of receive antennas of the AP are fixed. However, it should be noted that both the position and orientation of the array of transmitter antennas can vary within the stretcher, as they will depend on the medical operation itself and change over time.

As we are aiming at the characterization of the wireless communication channel through ray-tracing techniques, our goal is to identify the power and the delay of the different rays, so that the CIR can be obtained. Besides the line-of-sight (LoS) component, we consider four reflexive surfaces (the four walls of the room), as we are taking into account only the first-order reflected rays. Under these conditions, there is a total of I=5 rays (for nomenclature purposes, we consider that the LoS components have index i=1 and the other four rays have indexes from i=2 to i=5) between each pair of antennas. Note that the poor scattering environment of the CathLab can be explained by the existence of few scatterers. In addition, many of these potential scatterers can be composed by materials that mostly absorb the signals instead of reflecting them and are neglected in our analysis.

Let us define the position of a given transmit antenna as $a_{tx}^{\left(m\right)}=\left(x_{tx}^{\left(m\right)},y_{tx}^{\left(m\right)}\right)$, and the position of a given RX antenna as $a_{rx}^{\left(n\right)}=\left(x_{rx}^{\left(n\right)},y_{rx}^{\left(n\right)}\right)$. Unless otherwise stated, we considered linear antenna rays where the separation of RX antennas is $d_{N}=\lambda /2$ (where $\lambda =c/f_{c}$ is the carrier wavelength and *c* is the speed of light) and the spacing between the TX antennas is $d_{M}=\lambda /2$. As mentioned before, the RX array is fixed, and the position of the TX array can vary within (or near by) the stretcher. The position of the TX antennas lie in a portion of a circle with radius $d_{MN}$. The center of this circle is the position of one of the RX antennas. Figure 3 shows a set of different CathLab scenarios (i.e., different channel realizations) considering a 4×4 MIMO system and different values of $d_{MN}$. For a given channel realization, both the position and the orientation of the TX array can vary.

For obtaining the powers and delays of the *I* rays of each one of the M×N channels, we start by calculating the length of each ray. For this purpose, we considered the images method. The *i*th image of a given transmit antenna (i.e., the image associated to the *i*th ray) is denoted as $a_{tx,i}^{\left(n\right)}=\left(x_{tx,i}^{\left(n\right)},y_{tx,i}^{\left(n\right)}\right)$. Therefore, the length of the *i*th ray between the *m*th transmit antenna and the *n*th receive antenna can be calculated as
(1)$$R_{i}^{\left(n,m\right)}=\left|\left|a_{tx,i}^{\left(m\right)}-a_{rx}^{\left(n\right)}\right|\right|=\sqrt{{\left(x_{tx,i}^{\left(m\right)}-x_{rx}^{\left(n\right)}\right)}^{2}+{\left(y_{tx,i}^{\left(m\right)}-y_{rx}^{\left(n\right)}\right)}^{2}}.$$

Figure 4 shows an example of the application of the images method applied to the CathLab environment, considering a MIMO system with M=2 and N=2 (for demonstration purposes, only the images associated to one pair of antennas are shown).

Clearly, in addition to (Equation 1), the length of the *i*th ray between the *m*th transmit antenna and the *n*th receive antenna is also given by $R_{i}^{\left(m,n\right)}=\tau_{i}^{\left(m,n\right)}c$. This means that the delay associated to the *i*th ray of the channel between the *m*th transmit antenna and *n*th receive antenna is
(2)$$\tau_{i}^{\left(n,m\right)}=\frac{R_{i}^{\left(n,m\right)}}{c}.$$

Let us now assume that the power of the signals transmitted by each antenna is $P_{TX}$, regardless of the MIMO size. In this condition, the power received in the *i*th ray, associated with the *n*th receive antenna and the *n*th transmit antenna, is given by
(3)$$P_{RX,i}^{\left(n,m\right)}={\left(\frac{\lambda }{4\pi R_{i}^{\left(n,m\right)}}\right)}^{2}P_{TX}G_{TX}G_{RX},$$
where $G_{TX}$ and $G_{RX}$ are the gains of the transmit and receive antennas (which we are assuming to be unitary as we are considering isotropic antennas). The total power associated to a given receive antenna is given as the sum of the power of the *I* received multipath components, i.e.,
(4)$$P_{RX}^{\left(n\right)}=\sum_{i=1}^{I}P_{RX,i}^{\left(n,m\right)}.$$

Figure 5 shows the contour of the received power along the room, considering a single-input, single output (SISO) system (i.e., M=N=1) and different positions of the TX antenna. In this contour plot, the received power remains constant in a given oval as long as the TX is positioned in that oval.

As expected, the received power will be higher in the region occupied by the stretcher. Let us now see what is the impact of neglecting the LoS component. This situation can occur if there is a blockage between the TX and the RX. Figure 6 shows the contour of the received power along the room considering the same conditions of the Figure 5, with the exception of neglecting the LoS component.

Clearly, the received power will be lower. Regarding the stretcher, where it is likely that the TX is transmitting, the power can be 15 dB lower than when the LoS component exists. Moreover, note that when there is not a LoS component, the received power will be higher near the reflexive surface that is closer to the RX.

Let us now define the LoS rays as
(5)$$\alpha_{1}^{\left(n,m\right)}=\sqrt{P_{1}^{\left(n,m\right)}}\exp\left(-j2\pi \frac{R_{1}^{\left(n,m\right)}}{\lambda }\right).$$

For non-LOS rays, the reflection coefficient should be taken into account and the rays are generically given by
(6)$$\alpha_{i}^{\left(n,m\right)}=\sqrt{P_{i}^{\left(n,m\right)}}\exp\left(-j2\pi \frac{R_{i}^{\left(n,m\right)}}{\lambda }\right)\Gamma_{i},\;i>1,$$
where $\Gamma_{i}$ represents the reflection coefficient of the *i*th reflexive surface, which depends not only on the the material that composes the surface, but also on the carrier frequency [17]. Unless it is otherwise stated, we consider that $\Gamma_{i}=1\;\forall \;i$ for the *I* active multipath components.

By having the lengths and delays of each ray, we have sufficient information to compute the channel impulse response (CIR) between the *m*th transmit antenna and the *n*th receive antenna at a given time $t_{0}$, which we denote as $h^{\left(m,n\right)}\left(\tau ,t_{0}\right)$. In fact, note that the CIR is a function of both the delay-domain, τ, and time-domain, *t*. The time dependence of the CIR is explained by the variations on position of the transmitter (i.e., the PIM, held by the doctor), and not related to variations of environment, i.e., in the CathLab scenario it is reasonable to assume that the number and position of the scatterers are constant. Therefore, we consider that the the amplitude and delay of a given ray do not change if the positions of the RX and TX antennas are fixed. Generically, i.e., assuming that the position of the TX is varying over time, the CIR at a given instant $t_{0}$ can be defined as
(7)$$h{\left(\tau ,t_{0}\right)}^{\left(n,m\right)}=\sum_{i=1}^{I}\alpha_{i}^{\left(n,m\right)}\left(t_{0}\right)\delta \left(\tau -\tau_{i}^{\left(n,m\right)}\left(t_{0}\right)\right).$$

Regarding the frequency-domain, the channel frequency response is given by the Fourier transform of (Equation 7), which yields
(8)$$H{\left(f,t_{0}\right)}^{\left(n,m\right)}=\sum_{i=1}^{I}\alpha_{i}^{\left(n,m\right)}\left(t_{0}\right)\exp\left(-2\pi f\tau_{i}^{\left(n,m\right)}\left(t_{0}\right)\right).$$

Figure 7 shows a set of CIRs between a given pair of antennas considering a MIMO system with M=N=4 and $d_{MN}=0.6$ m.

The average delay spread of the CathLab, which weights the different delays by their relative power, can be calculated for a given channel realization obtained at $t=t_{0}$ as
(9)$${\bar{\tau }}^{\left(n,m\right)}\left(t_{0}\right)=\frac{\sum_{i=1}^{I}{\left|\alpha_{i}^{\left(n,m\right)}\left(t_{0}\right)\right|}^{2}\tau_{i}^{\left(n,m\right)}\left(t_{0}\right)}{\sum_{i=1}^{I}{\left|\alpha_{i}^{\left(n,m\right)}\left(t_{0}\right)\right|}^{2}}.$$

However, a better measure of the dispersiveness of the channel is the root mean square (RMS) delay spread, which, for a given channel realization, is given by
(10)$$\sigma_{\tau }\left(t\right)=\sqrt{\frac{\sum_{i=1}^{I}{\left|\alpha_{i}^{\left(n,m\right)}\left(t\right)\right|}^{2}{\left(\tau_{i}^{\left(n,m\right)}\left(t_{0}\right)-{\bar{\tau }}^{\left(n,m\right)}\left(t_{0}\right)\right)}^{2}}{\sum_{i=1}^{I}{\left|\alpha_{i}^{\left(n,m\right)}\left(t_{0}\right)\right|}^{2}}}.$$

Figure 8 shows the complementary cumulative distribution function (CCDF) of the RMS delay spread considering different distances between the transmitter and receiver.

From the figure, it can be noted that the for the distances of interest, i.e., within or near by the stretcher, the delay spread varies from 3.5 ns to 5.5 ns.

### 3.2. CathLab-Based Channels

To have an insight on what could be the performance in environments based on the CathLab, we can consider variations of the channel derived through the ray-tracing techniques. Let us start by considering a channel where the phase of each ray has a random factor. In such scenario, the *i*th ray associated to the channel between the *m*th transmit antenna and the *n*th receive antenna is given by
(11)$$\alpha_{i}^{\left(n,m\right)}=\sqrt{P_{i}^{\left(n,m\right)}}\exp\left(-j2\pi \frac{R_{i}^{\left(n,m\right)}}{\lambda }\theta_{i}^{\left(n,m\right)}\right)\Gamma_{i},$$
where $\theta_{i}^{\left(n,m\right)}$ is a random phase factor distributed uniformly in the range [0,2π]. Another possible variation is a CathLab channel, where the non-LOS rays (i.e., reflected rays) have a Rayleigh fading component. The rays of this channel can be defined as
$$u_{i}^{\left(n,m\right)}=\left\{\begin{array}{cc} \alpha_{i}^{\left(n,m\right)}, & i=0\\ \alpha_{i}^{\left(n,m\right)}\beta_{i}^{\left(n,m\right)}\Gamma_{i}, & i>0, \end{array}\right.$$
where $\beta_{i}^{\left(n,m\right)}$ is the Rayleigh fading component associated to the *i*th ray, *m*th transmit antenna, and *n*th receive antenna. This Rayleigh fading component is defined as $\beta_{i}^{\left(n,m\right)}∼\mathrm{Rayleigh}\left(\frac{1}{\sqrt{2}}\right)$, which means that $E[|\beta_{i}^{\left(n,m\right)}{|}^{2}]=1$, i.e., it has unitary average power.

## 4. Channel Capacity

In this section, we evaluate the channel capacity of an IEEE 802.11ax system operating in the CathLab environment. We consider a MIMO-OFDM system with *M* transmit antennas and *N* receive antennas, where the OFDM signals have $N_{s}$ subcarriers. The data symbols associated to a given subcarrier form the M×1 vector $S_{k}={\left[S_{k}^{\left(1\right)}\;S_{k}^{\left(2\right)}\;\cdots \;S_{k}^{\left(M\right)}\right]}^{T}$. The average symbol energy is denoted as $E[|S_{k}^{\left(m\right)}\left|\right]=2\sigma_{S}^{2}$. The cyclic prefix (CP) has duration 0.8 μs (the minimum value of IEEE 802.11ax), which is substantially higher than the RMS delay spread of the CathLab (see Figure 8).

By taking into account (Equation 8) and a given channel realization, (i.e., a given position of the TX array), the channel frequency response associated to the *k*th subcarrier and the pair of antennas (n,m) is $H_{k}^{\left(n,m\right)}≜H{\left(k/T_{s}\right)}^{\left(n,m\right)}$, where $T_{s}$ is the OFDM’s symbol duration. Under these conditions, we represent the channel associated to the *k*th subcarrier by the N×M matrix $H_{k}$, which is defined as
(12)$$H_{k}=\left[\begin{array}{cccc} H_{k}^{\left(1,1\right)} & H_{k}^{\left(1,2\right)} & \cdots & H_{k}^{\left(1,M\right)}\\ H_{k}^{\left(2,1\right)} & H_{k}^{\left(2,2\right)} & \cdots & H_{k}^{\left(2,M\right)}\\ \vdots & \vdots & \ddots & \vdots \\ H_{k}^{\left(N,1\right)} & H_{k}^{\left(N,2\right)} & \cdots & H_{k}^{\left(N,M\right)} \end{array}\right].$$

By considering (Equation 12), we can express the N×1 received signal associated to a given subcarrier by
(13)$$Y_{k}=H_{k}S_{k}+N_{k},$$
where $S_{k}$ and $N_{k}$ are the transmitted signal and noise components associated to the *k*th subcarrier, respectively. The noise component associated to a given subcarrier and receive antenna is modeled by a complex Gaussian random variable with variance $E[|N_{k}^{\left(n\right)}\left|\right]=2\sigma_{N}^{2}$. We define the ratio between the power of the useful signal and the power of the noise (i.e., the signal-to-noise ratio (SNR)) as
(14)$$\mathrm{SNR}=\frac{E[|S_{k}^{\left(m\right)}|]}{E[|N_{k}^{\left(n\right)}|]}=\frac{\sigma_{S}^{2}}{\sigma_{N}^{2}}.$$

For a given channel realization and subcarrier, the instantaneous channel capacity, which is calculated under the assumption that no channel state information (CSI) is available at the TX, can be defined as [18]
(15)$$C_{k,H}=\log_{2}\left(\det\left(I_{N}+\frac{\mathrm{SNR}}{M}H_{k}H_{k}^{H}\right)\right),$$
where $I_{N}$ denotes an N×N identity matrix. For an OFDM system, the total capacity is measured as the sum of the capacities of the individual subcarriers
(16)$$C_{H}=\sum_{k=0}^{N-1}C_{k,H}=\sum_{k=0}^{N-1}\log_{2}\left(\det\left(I_{N}+\frac{\mathrm{SNR}}{M}H_{k}H_{k}^{H}\right)\right).$$

The previous equation is valid for a given channel realization. To obtain the average capacity (average over several channel realizations), one must calculate the ergodic capacity, which is given by
(17)$$C=E_{H_{k}}\left[C_{H}\right].$$

In the following, we present a set of performance results regarding the ergodic capacity of the considered IEEE 802.11ax system. Unless otherwise stated, it is assumed that the number of subcarriers is $N_{s}=512$ and that the distance between the TX array and the RX array is variable.

Figure 9 shows the ergodic system capacity considering different channels and a SISO-OFDM system (i.e., M=N=1).

As can be noted, the system capacity obtained in the CathLab can be higher than the one obtained in a theoretical Rayleigh channel. This can be explained by the fact that the CathLab channel has lower fading effects than the Rayleigh channel.

Let us now consider a MIMO-OFDM system. Figure 10 compares the ergodic system capacity of both CathLab and Rayleigh channels considering different values of *M* and *N*.

From the figure, it can be seen that although in the SISO case the CathLab presents a larger capacity than the one obtained in the Rayleigh channel; the situation is different when MIMO systems are considered. In fact, for all MIMO systems, the ergodic capacity in the Rayleigh case outperforms the one obtained in the CathLab. This is justified by the fact that, in Rayleigh channels, the fading in each antenna is assumed to be uncorrelated. Therefore, the probability that the channel matrix has small singular values is lower, i.e., the channel matrix has a higher probability of being well conditioned. On the contrary, due to the poor scattering environment observed in the CathLab, one can expect a considerable degree of correlation between the RX antennas. This means that the channel matrix can be ill-conditioned with a larger probability, which leads to a lower channel capacity.

Clearly, an increase in the capacity of the CathLab involves a decorrelation of the channels on each receive antenna. This can be made by increasing the spacing between RX antennas, i.e., increase $d_{M}$. Figure 11 presents the ergodic channel capacity of a 4×4 MIMO system considering both a Rayleigh and a CathLab channel and different antenna spacings in the RX array.

From the figure, note that by increasing the spacing between the RX antennas, the capacity of the CathLab system increases and becomes closer to that the one obtained in uncorrelated Rayleigh channels. However, also note that at $f_{c}=5.1$ GHz, we have λ≈6 cm, which means that increasing $d_{N}$ above it (i.e., $d_{N}>\lambda$) might be not feasible, especially for MIMO systems with more than two receive antennas, as the RX array would become too large.

Let us now see what is the impact of considering the CathLab-based channels introduced in the Section 3.2. Figure 12 shows the ergodic capacity of an 8×8 MIMO-OFDM system considering both Rayleigh as well as different CathLab channels.

From the figure, note that when the CathLab channel is combined with a fading component on the non-LoS rays, the system capacity increases substantially relatively to the case where the pure CathLab channel is considered. Moreover, with a CathLab channel where a random phase factor is introduced on each ray, the capacity becomes equal to the one obtained in the ideal Rayleigh fading channel. This can be explained by the fact that both these variations of the CathLab channel leads to uncorrelated elements in the channel matrix, making it well-conditioned with a high probability and, consequently, improving the channel capacity.

## 5. Conclusions

In this work, we derive the channel model of the CathLab through ray-tracing techniques. This allowed us to study the channel capacity of a MIMO-OFDM scheme associated to an IEEE 802.11ax WLAN system. It is shown that in the CathLab channel, the MIMO gains are not as high as they would be if the channel matrix presents more uncorrelated elements (such as in Rayleigh fading channels), which can be explained by the poor scattering environment. Although increasing the spacing between antenna elements could reduce the correlation between antennas, it would also lead to very large antenna arrays, which might not be feasible. This means that increasing the number of antennas of the MIMO system might not be enough for achieving the very large data rates required for wireless medical imaging applications in general and for the IVUS catheter in particular. As obtaining a large bandwidth can also be very challenging due to the competitive nature of the medium access of the IEEE 802.11ax channels, the required data rates and QoS required for the IVUS will likely be obtained through a combination of MIMO schemes with a rigorous MAC design that enables the use of a stable, large bandwidth. 

## Figures and Tables

**Figure 1 sensors-20-00938-f001:**
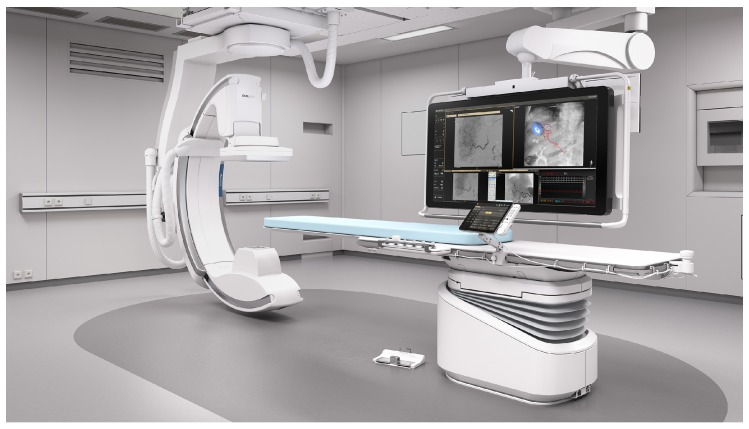
Example of a medical imaging laboratory (Philips CathLab^®^).

**Figure 2 sensors-20-00938-f002:**
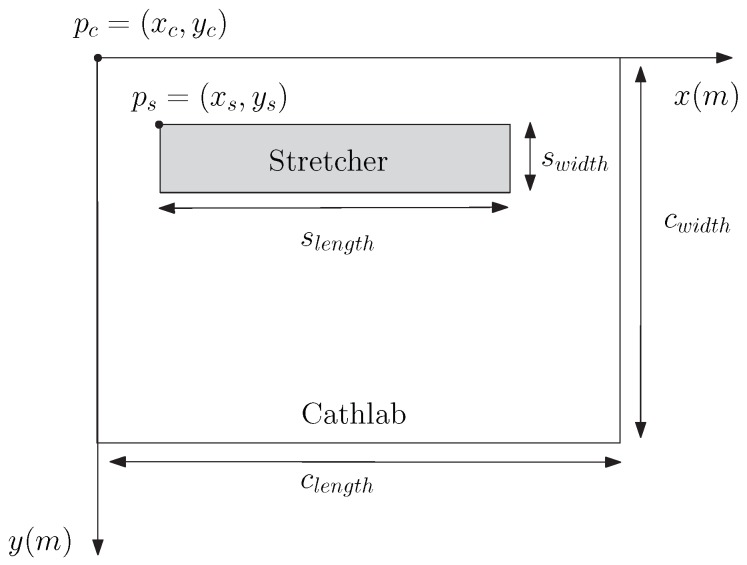
Top view of the CathLab: room and stretcher dimensions.

**Figure 3 sensors-20-00938-f003:**
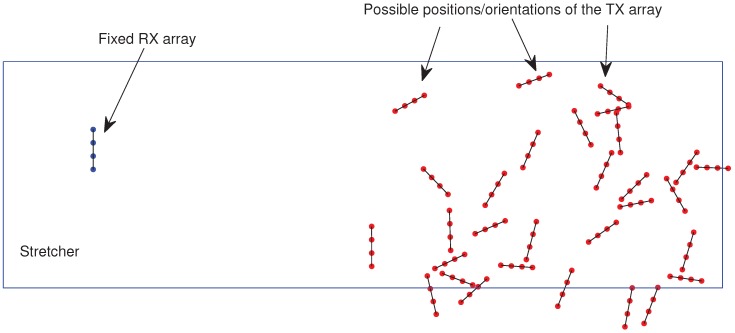
Different CathLab scenarios considering a 4×4 MIMO system and different values of $d_{MN}$.

**Figure 4 sensors-20-00938-f004:**
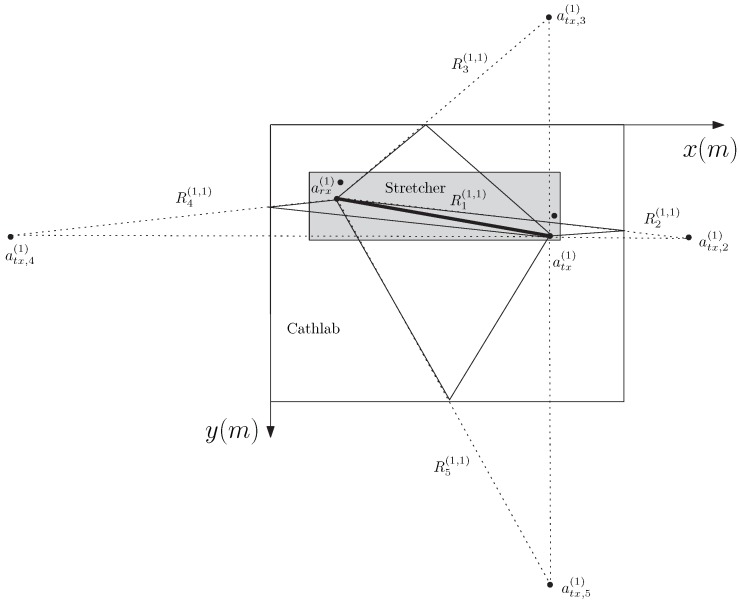
Example of the images method applied to the CathLab environment considering a MIMO system with M=2 and N=2.

**Figure 5 sensors-20-00938-f005:**
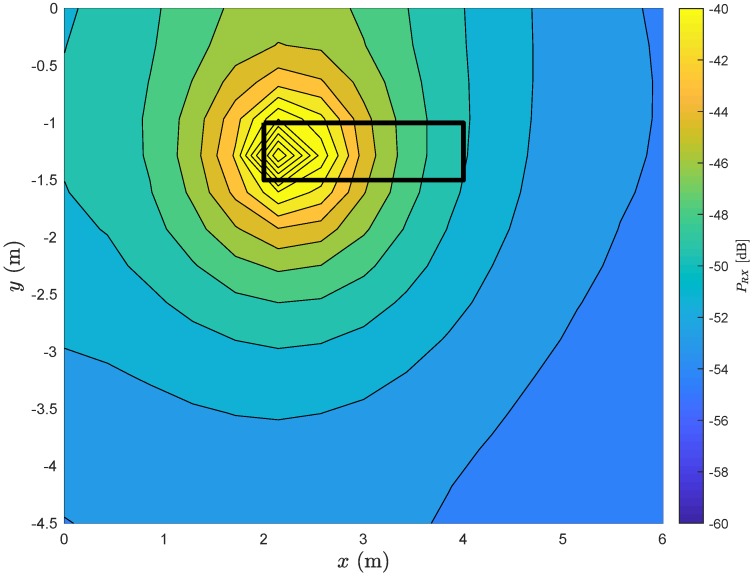
Contour of received power along the room considering a SISO system (i.e., M=N=1).

**Figure 6 sensors-20-00938-f006:**
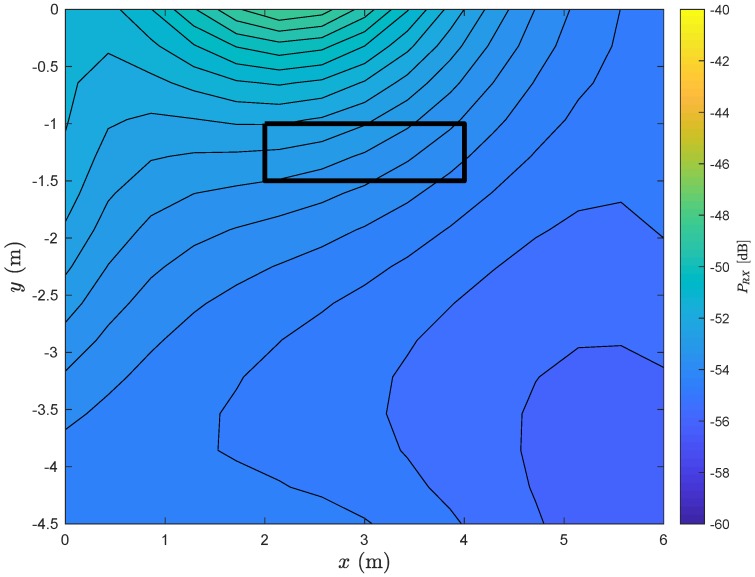
Contour of received power along the room considering a SISO system without the line-of-sight (LoS) component.

**Figure 7 sensors-20-00938-f007:**
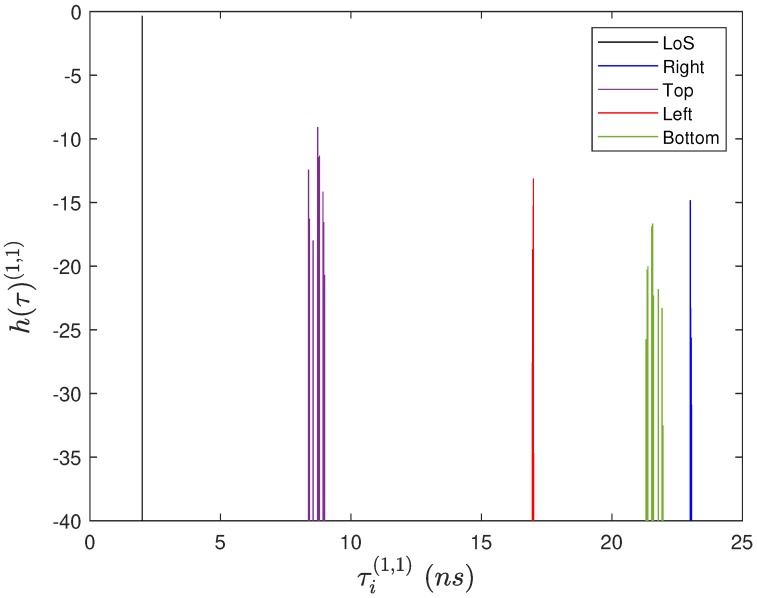
Set of channel impulse responses (CIRs) between the pair of antennas (1,1), considering a MIMO system with M=N=4 and $d_{MN}=1.0$ m.

**Figure 8 sensors-20-00938-f008:**
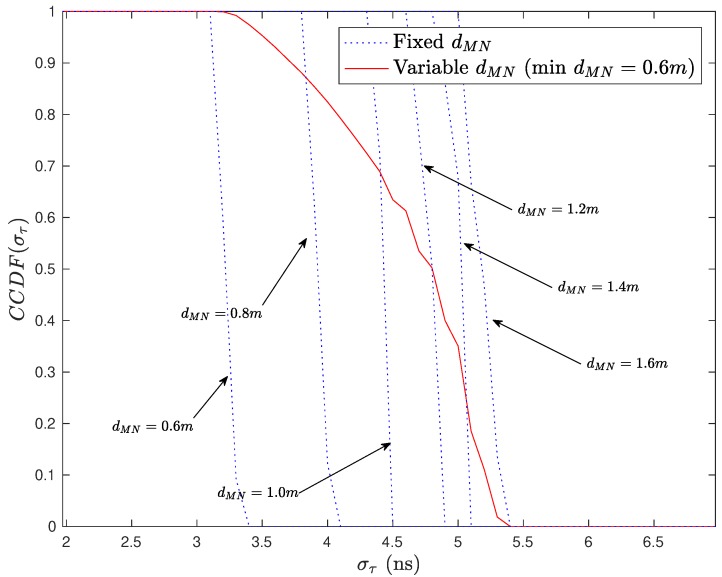
CCDF of the RMS delay spread considering different distances between the transmitter and receiver.

**Figure 9 sensors-20-00938-f009:**
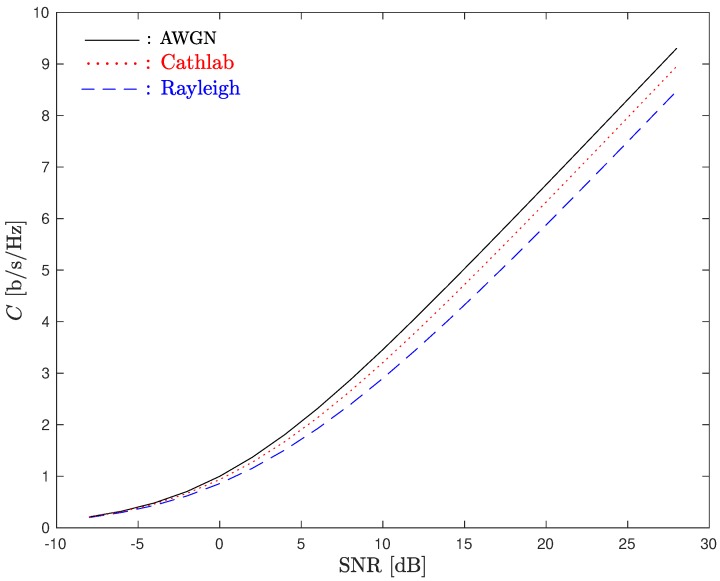
Ergodic system capacity considering different channels and a SISO-OFDM system.

**Figure 10 sensors-20-00938-f010:**
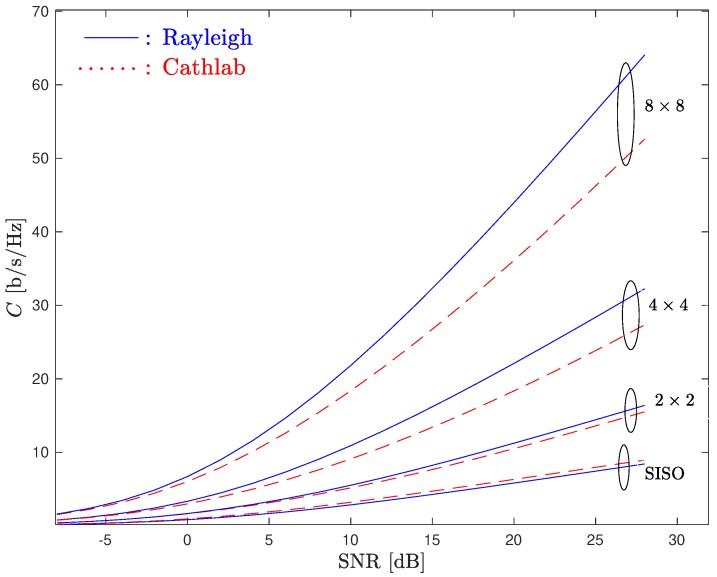
Ergodic system capacity for both Rayleigh and CathLab channels considering MIMO systems with different size.

**Figure 11 sensors-20-00938-f011:**
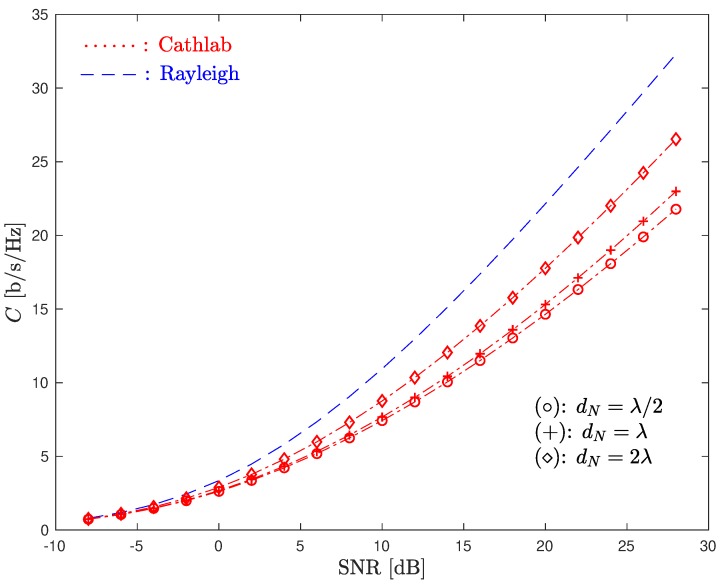
Ergodic system capacity with Rayleigh and CathLab channels with different values of $d_{N}$.

**Figure 12 sensors-20-00938-f012:**
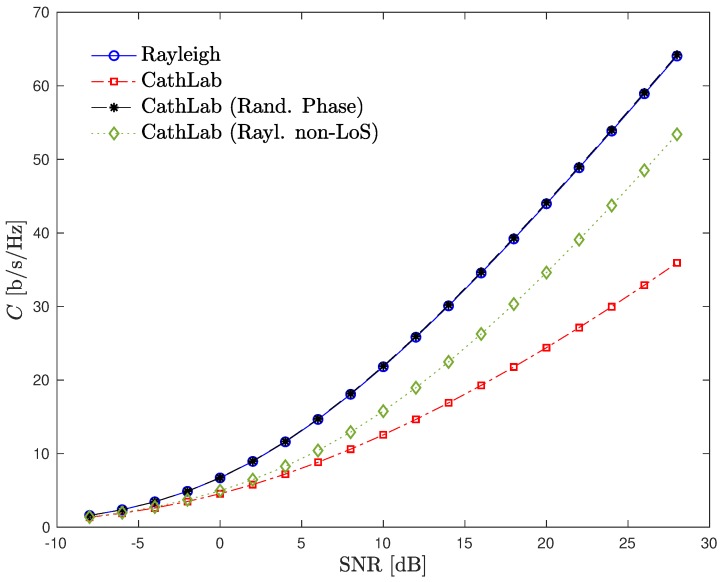
Ergodic capacity of an 8×8 MIMO-OFDM system considering different CathLab channels.

**Table 1 sensors-20-00938-t001:** Theoretical achievable rates for one SS considering different modulation and coding scheme (MCS) and different values of *B*.

MCS	Constellation	CR	*R* (B=40 MHz) [Mbps]	*R* (B=80 MHz) [Mbps]	*R* (B=160 MHz) [Mbps]
3	16-QAM	1/2	68.8	144.1	288.2
5	64-QAM	2/3	137.6	288.2	576.4
8	256-QAM	3/4	206.5	434.2	868.4
10	1024-QAM	3/4	258.1	540.4	1080.8

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
