# Peer review of "On the Achievable Capacity of MIMO-OFDM Systems in the CathLab Environment"

_sensors, 2020, doi:10.3390/s20030938_

Round 1
Reviewer 1 Report
Line91: “the stretcher position ps=(2,-1)m” which is out of the room according to the figure2. Please check it. The parameters that were used in the simulation should be given the specific figures, like Γi in Line 111. Line89: Can the authors give some explanation why the simplicity could be valid? Usually the height of the room is 3m that is comparable to the length and width.Author Response
Please see the attachment

Reviewer 2 Report
This manuscript reports a study based on the propagation scenario of a typical medical laboratory through ray- tracing techniques and they studied the potential of multiple-input multiple-output techniques in advanced WiFi 6 or wireless local area network (WLAN) communication standard IEEE 802.11ax settings. Based on the procedure, they have concluded that using MIMO techniques with a large 46 number of antennas might not be a good solution for achieving very high data rates in WiFi 6 settings. The manuscript is well written and I strongly recommend publication of this manuscript.
Reviewer 3 Report
The paper is an interesting study of the limitations of using IEEE 802.11ax standard as MIMO-OFDM system in the CathLab environment. The study is done using ray-tracing simulation techniques, and by evaluating the CathLab channel gains in comparison to theoretical maximum. In general, the paper is written with high detail, and it is rather easy to follow the motivation of the calculations. However, some fragmentation in the structure, most likely due to several authors, can be found. In addition, more explanations about the figures could improve the manuscript notably. In addition, some language revision should be done, mainly by rewording or reorganizing the order of words for better clarity. The main problem with the document is the conclusions section. You state that with MIMO system, the high data rates that are required for the wireless medical imaging applications cannot be reached, but this is not shown anywhere in the text. Instead in Fig.12 you show that with CathLab you can reach the performance of Rayleigh channel. In the text it is mentioned that the size limitations might limit the use of large MIMO systems, but this is not repeated in the conclusions. Some other corrections and comments: Line 1: the evolution … has been harnessed Line 2-3: communications is repeated three times, consider revising the sentence Line3: facilitate the logistics of the work of …, I am not sure what you are trying to say here. Line 9: It is shown -> We show Line 14: is -> are allows -> allow Line 26: Either this->these, or requirements->requirement Line 42: It is shown -> as previously, this tense refers to some other research if you use active tense in the other places. Line 44: You mention poor scattering environment, but you don’t return to this later in the text. In what way is the scattering environment in the CathLab poor? Line 45: put ‘based on the WLAN IEEE 802.11ax standard’ from the end after the word techniques Line 52: word ‘considering’ is a bit strange in this context, ‘looking at’ might be better in this Table1: Where do you get the values from? Is there a reference? Line 89: The height of room is much lower than its length. Is it comparable to the width? Figure 7: Consider using log scale on y, it is rather impossible to see the markers of the reflections Figure 10: You don’t mention the meaning of the ovals in the distributions.
Reviewer 4 Report
In lines 45-48 Authors anticipate conclusions reported in lines 168-170. I think that Conclusions should be slightly improved with further details. What Authors suggest to achieve "the very large data rates required for wireless medical imaging applications in general and for the IVUS catheter in particular"? Some sentences on these topics could strength the entire paper.
Round 2
Reviewer 1 Report
Line 116 wrote "Moreover, many of them are covered by materials
that mostly absorb the signals instead of reflecting them." while the
"reflection coefficient" gamma is defined as 1 (Line 129) that means total
reflection. It's not self-consistent.
